# Dedicated Material Models of EN AW-7021 Alloy for Numerical Modeling of Industrial Extrusion of Profiles

**DOI:** 10.3390/ma18133166

**Published:** 2025-07-03

**Authors:** Konrad Błażej Laber, Jacek Madura, Dariusz Leśniak, Maciej Balcerzak, Marek Bogusz

**Affiliations:** 1Department of Metallurgy and Metal Technology, Faculty of Production Engineering and Materials Technology, Czestochowa University of Technology, 19 Armii Krajowej Ave., 42-200 Częstochowa, Poland; 2Department of Materials Science and Non-Ferrous Metals Engineering, Faculty of Non-Ferrous Metals, AGH University of Science and Technology, Adama Mickiewicza 30 Ave., 30-059 Krakow, Poland; madura@agh.edu.pl (J.M.); dlesniak@agh.edu.pl (D.L.); balcerzak@agh.edu.pl (M.B.); bogusz@agh.edu.pl (M.B.)

**Keywords:** EN AW-7021 alloy, aluminum extrusion, porthole dies, rheological properties, plastometric testing, flow stress, numerical modeling

## Abstract

In this paper, dedicated material models were developed and verified for three melts of EN AW-7021 alloy, differing in zinc and magnesium content, for tube extrusion conditions. Based on the plastometric tests, it was found that in the studied range of strain parameters, the analyzed melts of the same aluminum alloy showed different sensitivity to strain rate and temperature. In addition, a significant effect of magnesium and zinc content on the plasticity of the tested material was observed. Therefore, dedicated material models describing stress changes were developed for each melt analyzed. The models were then implemented into the material database of the QForm-Extrusion^®^ program, which was used for the theoretical analysis of the industrial extrusion process. In order to verify the results of numerical calculations, industrial tests of the extrusion process were carried out. The force parameters and the rate of the extrusion process were mainly analyzed. The use of dedicated material models for each melt contributed to the accuracy of numerical modeling. A high degree of compliance was obtained regarding the theoretical and experimental extrusion force and the velocity of metal flowing out of the die cavity, among others.

## 1. Introduction

In order to properly design or modify existing technological processes, it is necessary to know the characteristics that describe the rheological properties of the studied material [1,2]. For each technological process, a set of characteristics can be determined that describe the plastic forming material’s susceptibility in a given process. In the case of plastic forming processes, the essential characteristics that determine the ability of a given material to be shaped plastically are the flow stress ($\sigma_{P}$) and the limiting strain ($\varepsilon_{L}$) [2]. The flow stress $\sigma_{P}$, i.e., the stress required to initiate and continue the plastic flow of a metal under uniaxial stress conditions, is a function of strain (ε), strain rate $\left(\dot{\varepsilon }\right)$, temperature (T) and strain history [2]. Determination of the technological plasticity of materials is particularly difficult for hot plastic processing conditions, since in the structure of the material there are simultaneously processes of strengthening resulting from the mechanism of plastic strain, the presence of particles of foreign phases in solution and in the form of precipitates, and heat-activated softening processes that lead to weakening of the material [2]. The correct determination of the rheological properties of the analyzed material in the form of stress–strain diagrams, taking into account the effect of material temperature and strain rate, increases the accuracy of calculations when using empirical formulas, as well as during numerical modeling using the finite element method (FEM) [3,4].

The course of the strengthening curves of the tested materials and the value of flow stress can vary significantly, depending on the test method (compression, tensile or torsion test). This difference is particularly noticeable for metals and alloys with hexagonal lattices, as well as for materials with inhomogeneous structures when significant anisotropy of properties is present. Among the main factors that affect the results of plastometric testing are [5] the type of the material’s crystal lattice, the structural state and anisotropy of the property, the thermal effect due to plastic strain and the occurrence of a temperature gradient along the length and cross-section of the specimen, the method of clamping the specimen and the conditions at the contact surface of the specimen with the clamp, the effect of dynamic loading and stiffness of the “machine-sample” system (especially in tension), the formation of strain localization and the differences in strain rate along the length and cross-section of the specimen.

A detailed description of various test methods for performing plastometric tests and determining rheological properties is presented, among others, in [5,6,7]. In addition, these articles describe the research methodology and present the characteristics of various devices for plastometric testing. In addition, papers [5,7] present plasticity tests on selected steels, zinc alloys, magnesium alloys, titanium alloys, copper and nickel alloys, and alloys on the topic of intermetallic phases. Paper [8] discusses in detail the research methodology for determining plasticity using torsion testing. Numerous results of plastometric tests aimed at determining rheological properties and developing material models based on them, for numerical modeling of various types of plastic processing, are presented in articles [9,10,11,12,13,14,15,16,17,18,19,20,21,22,23], among others.

According to papers [9,10,11,12,13,14,15], determination of the flow stress of the studied material and the material model is important, among others, when analyzing, designing and optimizing forging processes [9,10]. Paper [11] analyzed the effect of the stress–strain curves of steel C45 at different strain rates on the flange forming force during tube production. In addition, optimization of the flange forming process was carried out through numerical simulation and experimental verification. The plastometric results presented in [12] were used for studying the evolution of the microstructure and behavior of the material during hot forging processes in the temperature range of 900–1200 °C and the strain rate range of 0.01–20 s^−1^. In addition, the strain-compensated Arrhenius constitutive relationship for 25CrMo4 steel was identified and the kinetics of microstructure evolution were determined, including dynamic recrystallization, metadynamic recrystallization, static recrystallization and grain growth. A coupled numerical model was then created for forging axes from the 25CrMo4 steel using FORGE NXT 1.1 ^®^ software. According to paper [13], the plastometric tests carried out on the AA6351 aluminum alloy and the material model developed on the basis of these tests enabled an accurate numerical analysis of the process of forging tubular and cylindrical components. The numerical modeling results were then verified with a laboratory experiment. Article [14], on the other hand, presents a procedure for determining and obtaining a new constitutive model for characterizing the flow stresses of 42CrMo4 steel in the temperature range from 1250 °C to 1375 °C. This contributed to discovery of the models of the material’s flow stress behavior at high temperatures and made it possible to predict or simulate hot forging processes at high temperatures in closed dies. In turn, the plastometric tests and the developed material model of C45 steel presented in [15] were used for developing and implementing predictive control in the hot forging process in open dies of heavy, large and hard deformable steel forgings. Based on the analysis of the test results obtained, the predicted hot flow stresses and predicted strain/forging forces were determined, among others. The predicted parameters of the forging process were then incorporated into the dynamic model of the hydraulic forging press.

The correct determination of the flow stress of the studied material and the material model is also important when designing extrusion processes, as confirmed by the test results published in papers [16,17,18,19], among others. The results of plastometric tests and the developed material model presented in [16] were used for numerical analysis of the skin contamination behavior in the extrusion of an AA6082 profile. Moreover, an innovative method for skin contamination prediction in the extrusion process was presented and discussed. In turn, the results of plastometric tests carried out within the framework of paper [17] made it possible to build a material model of aluminum alloy AA6005A and then develop (using numerical modeling) the technology for extruding a new product using an innovative die. In contrast, carrying out plastometric tests on the EN AW-1080A aluminum alloy and CW004A copper alloy, presented in paper [18], enabled the authors to carry out numerical modeling and analysis of the most important parameters of the process of indirect extrusion of copper-clad aluminum (CCA) bars and compare the obtained results with experimental studies. As a result, the development of extrusion force and equivalent copper cross-section could be clarified. Additionally, the validated numerical analysis made it possible to determine the conditions for a successful co-extrusion of the analyzed CCA rod. In turn, in paper [19], on the basis of the conducted plastometric tests and the developed material model, the authors proposed a new numerical scheme for calculating stress fields and metal flow rates in the axisymmetric steady-state extrusion process of Al 6351 and Al 6060 aluminum alloys, using the finite volume method (FVM).

Accurate determination of the flow stress of the studied material and the material model is also important when designing the rolling processes, as confirmed by the research results published, among others, in [20,21,22,23]. The plastometric studies presented in paper [20] and the material model of the tested steel developed on the basis of these studies were used for a numerical analysis of the possibility of influencing the residual stress state in rolled railroad rails by changing the design of the passage of vertical and horizontal straightening rollers and optimizing their distribution around the perimeter of the rail. A wide range of theoretical considerations based on the use of the finite element method using the commercial FORGE^®^ software package were carried out and verified under industrial conditions. In turn, in paper [21] plastometric tests were used to develop a material model that was used for predicting rolling force, among others. In turn, conducting plastometric tests on the AZ31 magnesium alloy, presented in paper [22], and the material model developed on the basis of these tests, enabled the authors to carry out a numerical analysis of the process of one-pass bar rolling in a three-high skew rolling mill. Based on the results obtained, the effect of the rolling speed on the torsion of the bar and the state of stress and strain occurring in the rolled strip was determined. Performing plastometric tests and developing a material model of the 38MnVS6 steel, in turn, enabled the authors of paper [23] to examine the effect of pass schedule and groove design on the metal strain of the investigated steel grade in the initial passes of hot rolling.

Each plastometric test (compression, tension, torsion) has certain advantages and disadvantages. The use of a given test method should result from an accurate reproduction or approximation of the actual conditions occurring in the plastic processing being analyzed [5]. The method of swelling cylindrical and flat specimens is most versatile in terms of the closeness of the strain pattern to many metal forming processes and can be used in structural testing under hot strain conditions [5].

Plastometric tests, complemented by numerical modeling using the finite element method (FEM), can also be successfully used for characterizing the strain behavior of metallic materials. However, the accuracy of simulations that predict the strain behavior of materials is primarily dependent on the rheology law used. This is confirmed by the tests presented in paper [24,25], among others.

The accuracy of the mathematical model describing changes in the flow stress value depending on the strain parameters is particularly important for processes characterized by high strain values, such as extrusion or torsion. This is confirmed by the test results published in papers [26,27], among others.

In this paper, dedicated material models were developed and verified for three melts of the EN AW-7021 alloy [28], differing in zinc and magnesium content, for conditions characteristic of the extrusion process of tubes with a diameter of 50 mm and a wall thickness of 2 mm. On the basis of preliminary plastometric studies of the EN AW-7021 alloy, it was found that in the studied range of strain parameters, the three melts analyzed showed varying sensitivity to strain rate and temperature, depending on their main alloying element contents. In addition, a significant effect of magnesium and zinc content on the flow stress and plasticity of the material was observed. Accordingly, three dedicated material models (mathematical models) were developed to describe the changes in flow stress as a function of strain, strain rate and temperature, for each melt analyzed. In the available technical literature, no papers were found that demonstrate in detail the effect of the content of the main alloying elements (Zn and Mg) on the value of the flow stress and plasticity of the EN AW-7021 aluminum alloy.

The novelty of the paper compared to previously published research results is the use of (various) dedicated material models for the same grade of aluminum alloy (EN AW-7021). Based on the tests conducted, the approach used made it possible to increase the accuracy of the numerical modeling of the extrusion process analyzed—which was confirmed by industrial tests.

## 2. Materials and Methods

### 2.1. Materials

The tests presented in this paper were carried out for 3 melts (billets) of aluminum alloy of the EN AW-7021 grade [28], differing in zinc and magnesium content, with the chemical composition shown in Table 1. The billets were cast under semi-industrial conditions and then subjected to a dedicated homogenization process.

### 2.2. Methods

The tests presented in the paper were conducted in several stages. In the first one, plastometric tests were conducted on three analyzed aluminum alloy melts of the EN AW-7021 grade, differing in zinc and magnesium content. These tests were carried out for conditions characteristic of the extrusion process of tubes with a diameter of 50 mm and wall thickness of 2 mm. In the second stage, the results of the performed plastometric tests were approximated by the Hensel–Spittel equation [1], to determine its coefficients. In the next stage of the tests, the developed mathematical models of each melt were then implemented into the material database of the QForm UK Extrusion^®^ program [30], which was used for the theoretical analysis of the industrial extrusion process. In the final stage, industrial verification of the extrusion process was carried out on a 25 MN hydraulic press with a 7-inch-diameter container. The force parameters and rate of the extrusion process were mainly analyzed.

#### 2.2.1. Plastometric Test Methodology

Plastometric tests of the three analyzed aluminum alloy melts of the EN AW-7021 grade, on the basis of which their rheological property characteristics were developed and the stress function coefficients were selected, were carried out in uniaxial compression testing using the GLEEBLE 3800 simulator (Dynamic Systems Inc., Poestenkill, NY, USA) [31]. The compression test was performed using a special measuring unit that is installed in the Hydrawedge system [32,33]. The special structure of the test head and the use of two independent hydraulic systems made it possible to skip the acceleration and deceleration phases of the shaping tool’s movement during specimen straining. The technical solutions used in the Hydrawedge system make it possible to reduce the temperature difference along the length of the sample to less than 3 °C [32]. In addition, the Hydrawedge system allows the programmed experiment to be executed very accurately [32]. The anvil, with the help of which the strain is inflicted, obtains the preset strain rate in a very short time and maintains it until the strain is completed, maintaining the exact preset value of partial and total strains, even at very high strain rates [32,33].

To determine the true strain ε, strain rate $\dot{\varepsilon }$ and flow stress $\sigma_{P}$, during compression, the following relationships were used [5]:(1)$$\varepsilon =\left|ln\frac{h_{1}}{h_{0}}\right|$$(2)$$\dot{\varepsilon }=ln\frac{\varepsilon }{∆t}$$(3)$$\sigma_{P}=\frac{4\cdot F\cdot h}{h_{0}\cdot \pi \cdot d_{0}^{2}}$$
where $h_{0}$ and ${\;h}_{1}$—initial and final height of the specimen; ∆t—strain time; *F*—force value measured during the test; *h* –height of the specimen; $d_{0}$—initial diameter of the specimen.

Plastometric tests were planned so that the flow stress function and its coefficients could be developed for the strain process conditions characteristic of the extrusion process. Plastometric tests were conducted for the following parameters: temperature: 450 °C, 480 °C, 510 °C, 540 °C, 570 °C; strain rate: 0.05 s^−1^, 0.5 s^−1^, 5 s^−1^; true strain: 1.2.

The tests were conducted in a vacuum at a constant temperature of the deformed sample. Cylindrical specimens with s diameter of d = 10 mm and height of h = 12 mm were used for the tests. Graphite washers and a special graphite-based lubricant were inserted between the specimen faces and tool surfaces to minimize friction. Two K-type thermocouples (NiCr-NiAl), welded to the side surface of the specimen, were used for recording and controlling temperature changes. The samples were heated at a constant rate of 5 °C/s to a given temperature, maintained at this temperature for 10 s, and then deformed.

#### 2.2.2. Development of Dedicated Material Models

In the available computer programs for solving problems in the field of plastic flow of metal or for calculating force by the finite element method, the values of flow stress $\sigma_{P}$ depend on the adopted flow stress function (model). Most commonly, the flow stress is described by a relationship in the form of $\sigma_{P}=(\varepsilon ,\dot{\varepsilon },T)$. A number of mathematical functions (models) are used for describing mathematically the changes in $\sigma_{P}$ as a function of strain, temperature and strain rate, which can be found in papers [1,2,32,34,35,36], among others. In paper [1], 17 flow stress functions (models) were analyzed to find the relationship that most accurately represents the true metal flow curves, for hot plastic processing conditions.

In the present paper, the Hensel–Spittel equation, which can be transformed to the form of (4), was adopted to describe the changes in $\sigma_{P}$ values for the studied melts of the EN AW-7021 alloy. The relationship is often used for determining the value of $\sigma_{P}$ in computer programs for numerical modeling of plastic processing [3].(4)$$\sigma_{P}=A\cdot e^{m_{1}\cdot T}\cdot T^{m_{9}}\cdot \varepsilon^{m_{2}}\cdot e^{\frac{m_{4}}{\varepsilon }}\cdot {\left(1+\varepsilon \right)}^{m_{5}\cdot T}\cdot e^{m_{7}\cdot \varepsilon }\cdot {\dot{\varepsilon }}^{m_{3}}\cdot {\dot{\varepsilon }}^{m_{8}\cdot T}$$
where $\sigma_{P}$—flow stress; *T*—temperature; ε—true strain; $\dot{\varepsilon }$—strain rate; *A*, *m_1_*÷*m_9_*—coefficients.

The coefficients of Equation (4) were determined. The values of these coefficients are shown in Table 2, Table 3 and Table 4.

#### 2.2.3. Numerical Modeling of the Extrusion Process

##### Mathematical Models and Calculation Assumptions

Conducting the numerical simulations required a number of preparatory tasks, including development of a digital 3D CAD model of both the die itself and the entire toolset to enable installation in an industrial press. The three-dimensional tool models were developed using SolidWorks 2020 CAD/CAM software [37] (Dassault Systèmes, Vélizy-Villacoublay, France).

The preparatory work and the numerical simulations themselves were carried out using the QForm UK Extrusion 3D v.10.1.7 software (Micas Simulations Ltd., Oxford, UK). A special software module allows importing the CAD model of the tools and generating a finite element mesh on their surface and in the model volume.

The numerical model of simulations performed using the finite element method is defined based on the flow formula proposed by Zienkiewicz and Pittman where the deformed material is treated as an incompressible and rigid viscoplastic continuum, while elastic strains are neglected. Calculations are performed based on the Euler–Lagrange model, which uses finite elements to simultaneously relate material flow to the strain and temperature distribution of the tool. This means that the elastic strain of the die affects the way the metal flows, while the strain of the tool itself is determined by the pressure of the metal on its surface [30].

The software is based on two discrete models. The first, the Lagrange model is designed to simulate the transient state of the initial stage of the process when the metal fills the die, while the second, the combined Euler–Lagrange model, is designed to simulate at the steady-state stage. In the first stage, the finite element mesh moves with the flowing metal to accurately represent the filling progress of the die. In particular, in the case described within this paper, extrusion through a porthole die where the metal separates at the bridges supporting the core and rejoins in the welding chambers, the Lagrange model determines in great detail how the material flows. This model is effective until the die clearance is filled, where the metal flow is less intense. In the steady state of the process—understood in the context of filling of the die and the uniform decrease in the extrusion force after the maximum value is reached—where it is important to determine the parameters of the flow of metal out of the die opening, a combined Lagrange–Euler model based on the assumption of complete and invariant filling of the interior of the die is used. This means that the finite element mesh inside the tool represents the computational domain of the space to be simulated. The generated finite element mesh is stationary, while the material flows through it according to the defined parameters during the simulation. Therefore, practically speaking, the program does not re-create the domain element mesh inside the tool but performs calculations in the already defined nodes. The results obtained at this stage of the simulation are the basis for defining the uniformity of the outflow of the press leaving the bearing. Due to the uneven manner in which the metal flows into the die cavity—determined, among others, by the structure or parameters of the process—the final geometry of the cross-section of the extruded section may be distorted. The purpose of running the simulation is to identify and define this undesirable effect.

One of the most important parameters that translates into the quality of the results obtained is the characteristics of the plastic flow of the metal and the flow stress for varying temperature values. The developed discrete material model in the form of the Hensel–Spittel constitutive equation was implemented in the FEM program database. This enabled numerical simulations of the extrusion process using the designed die set for selected aluminum melts.

Conducting numerical simulations made it possible to determine the change in the value of the punch force on the billet face during the extrusion process. Based on the obtained values, the dependence of the change in extrusion force as a function of punch displacement was plotted. On this basis, in the first stage of the work, under the conditions of numerical simulations, the selection was made of the optimal melt and material model for further testing under industrial conditions. At a further stage, the plotted relationships were compared with the data recorded during actual extrusion tests, which allowed direct verification of correctness of the numerical calculations.

The extrusion process was carried out using an innovative proprietary two-cavity porthole die adapted to an industrial extrusion press along with the toolset. An overview design of the die is shown in Figure 1.

##### Simulation Conditions and Parameters

Numerical simulations were carried out using high-quality 3D models—the computational domain (as the volume of aluminum filling the interior of the die) and the die set itself. To obtain high-quality results, a locally refined finite element mesh was used. High-density mesh element coefficients with a corresponding element size factor were used for zones where plastic strain is intensified. Particularly in the area of metal movement into the pre-chamber, a profile geometry adaptation factor of 0.45 was applied. An adaptation of 0.80 was applied to the calibration bearing, compared to an element size factor of 1.00 in the container itself. Figure 2 shows a mesh in the computational domain with a view of local densification. Figure 3 shows the volumetric and surface mesh in a porthole die.

Figure 3 shows the visualization of the surface and volume finite element mesh in the computational domain—die filling. The billet section and the die entry area show large elements with a size factor of 1.00. An increase in compaction and the number of elements can be observed in the direction of the main strain zone, especially in the area of strain intensification in the bearing area—where the elements have the smallest size and the density of nodes is the highest.

In the simulations carried out, the friction model defined by Levanov was applied at the contact surface between the deforming metal and the tool [38]:(5)$$F_{t}=m\cdot \frac{\sigma_{p}}{\sqrt{3}}\cdot \left(1-e^{\left(-1.25\cdot \frac{\sigma_{n}}{\sigma_{p}}\right)}\right)$$
where *m*—friction factor; $\sigma_{n}$—normal contact pressure; $\sigma_{p}$—flow stress. This equation can be considered a combination of the constant shear friction model and the Coulomb friction model. The second term in brackets in Equation (5) accounts for the influence of normal pressure at the contact interface. For high contact pressures, the expression approximates the conditions described by the constant shear model. In contrast, for low contact pressures, it defines a linear approximation dependent on the normal stress at the interface.

#### 2.2.4. Industrial Verification

Industrial tests were carried out in manufacturing conditions on a horizontal hydraulic press with a maximum load of 25 MN with a container sized Ø178 mm (7-inch) (manufactured in Italy—Extral Technology S.R.L., Cividino, Italy).

The press is equipped with a measurement and data acquisition system, which made it possible to record and analyze a number of process variables, including ones that are key for verifying the correctness of numerical simulations. The process was carried out according to the standard procedure of the industrial extrusion process on the selected press where a billet with a diameter of Ø178 mm and length of 800 mm was heated to 480 °C. The toolset was heated to 460 °C while the bolster was heated to 340 °C. The set punch velocity was 0.6 mm/s, which translated into a profile speed of 1.5 m/min.

The key in the supervised study was primarily to determine the pressure change in the hydraulic system of the press, which would enable the force characteristics of the extrusion process to be plotted. The measurement of pressure in the press system as a function of punch displacement is carried out through pressure sensors that constitute equipment of the press. This parameter is monitored while the process is running with the ability to read the course on the press control panel. In turn, the measurement of press speed and temperature is already performed on the press, using external equipment.

The second key indicator of the correctness of numerical simulations is the geometry of the obtained aluminum profiles. In practical terms, it is common in practice to compare the geometry of the initial part of the profile of the so-called “nose piece”. This allowed us to determine the characteristics and manner of metal flow out of the die cavity—for the purposes of this paper, the manner of metal flow was evaluated, including the unevenness associated with the manner of cavity placement and die design. This is important because the local measurement of velocity alone is not objective, as the specifically set values in the simulations are reflected in the measured puller speed. For this reason, in order to assess the subtleties involved in the specifics of metal flow in the die itself, it is important to identify and reliably evaluate the mode of metal flow in this initial portion of the extrudate. It is also a direct indicator of the correctness of the die structure. The extrudate material obtained was also subjected to geometric accuracy tests to validate the quality of the products.

#### 2.2.5. Geometry Measurements

Due to the need for precise and reproducible data on tube geometry, the measurements were conducted with an Atos Core 200 optical 3D scanner (Carl Zeiss GOM Metrology GmbH, Braunschweig, Germany), based on structured light technology. This device allows mapping of the objects studied in the form of three-dimensional models with very high accuracy, up to 0.017 mm. Using this technology, it was possible to reproduce digital spatial models of all the geometric variants of the tubes analyzed. Based on the 3D models created, a detailed evaluation of key geometric parameters, such as wall thickness and diameters (outer and inner), was carried out. The wall thickness measurement consisted of analyzing the distance between the outer and inner surfaces perpendicular to the surface at selected measurement points.

In order to determine the outer and inner diameters, cross-sections of the model were made, and then the circle was fitted to the resulting profile. Measurements were made in different cross-sections along the axis of the tube’s length, which made it possible to assess the dimensional uniformity along the entire length. Due to the limitations of the measurement method used, only areas with reasonably well-mapped geometry were analyzed for the inner diameter.

## 3. Results

### 3.1. Analysis of Plastometric Test Results and Development of Material Model Coefficients of Aluminum 7021-1

The true and approximated courses of changes in the flow stress of aluminum 7021-1, over the range of temperature and strain rate studied, are shown in Figure 4.

Based on the analysis of the experimental data (blank symbols) in Figure 4, it can be concluded that the studied aluminum alloy, i.e., 7021-1, had good deformability in the studied range of strain parameters. Within the tested range of temperatures, applied values and strain rates, the material retained its cohesion. After analyzing the course of true changes in the flow stress of the aluminum melt studied (blank symbols) for strain rates of 0.05 and 0.5 s^−1^, its value was found to increase with increasing strain over the entire temperature range studied.

After analyzing the course and nature of the true changes in the flow stress of the test melt (blank symbols) deformed at 5 s^−1^, it was found that after reaching the maximum value, the flow stress remained constant. This may indicate that healing processes were taking place in the studied material. Based on the authors’ previous experience, the slight increase in the value of flow stress observed at the end of the strain process at a speed of 5 s^−1^ may be due to the very nature of the compression test and may be caused by friction of the material against the tool surface, despite the use of graphite lubricant and graphite washers.

After analysis the effect of the strain rate of the test melt deformed at 450 °C on the increase in true values of flow stress (blank symbols), it was found that as the strain rate increased from 0.05 s^−1^ to 0.5 s^−1^, the flow stress increased (on average) by ca. 42%. In contrast, an increase in the strain rate from 0.5 s^−1^ to 5 s^−1^ resulted in an increase in the true flow stress values (on average) of about 16%.

Based on an analysis of the experimental data (blank symbols) shown in Figure 4b, it was found that as the strain rate increased from 0.05 s^−1^ to 0.5 s^−1^, the true flow stress (blank symbols) increased (on average) by about 39%. In contrast, an increase in the strain rate from 0.5 s^−1^ to 5 s^−1^ resulted in an increase in the true flow stress values (on average) of about 22%.

Based on an analysis of the effect of strain rate of aluminum 7021-1 deformed at 510 °C on the increase in true values of flow stress (blank symbols), it was found that as the strain rate increased from 0.05 s^−1^ to 0.5 s^−1^, the flow stress increased (on average) by ca. 48%, while an increase in strain rate from 0.5 s^−1^ to 5 s^−1^ resulted in an increase in true values of flow stress (on average) by about 42%.

The data in Figure 4d show that as the strain rate increased from 0.05 s^−1^ to 0.5 s^−1^, the true values of flow stress (blank symbols) increased (on average) by about 44%. In contrast, an increase in the strain rate from 0.5 s^−1^ to 5 s^−1^ resulted in an increase in the true flow stress values (on average) of about 41%.

Based on an analysis of the effect of strain rate of the tested material deformed at 570 °C on the increase in the true values of flow stress (blank symbols), it was found that as the strain rate increased from 0.05 s^−1^ to 0.5 s^−1^, the flow stress increased (on average) by ca. 50%, while an increase in strain rate from 0.5 s^−1^ to 5 s^−1^ resulted in an increase in true values of flow stress (on average) by about 62%.

After analysis of the effect of temperature (over the entire range studied) on the level of true values of the flow stress of aluminum 7021-1 (blank symbols) deformed at speeds of 0.05 s^−1^ and 0.5 s^−1^, it was found that in the temperature range of 450 °C–570 °C, the flow stress decreased (on average) by about 53%. When the test alloy was deformed at a rate of 5 s^−1^, the true values of flow stress were found to decrease (on average) by ca. 37% over the temperature range studied.

The course of the true and approximated plastic flow curves of aluminum 7021-1 in the studied range of strain parameters allowed us to observe a high correspondence between the true values of the flow stress and the values obtained by approximation.

Table 2 shows the values of the coefficients of the Hensel–Spittel Equation (4), which approximates the results of plastometric tests of aluminum 7021-1.

**Table 2 materials-18-03166-t002:** Material model coefficients of aluminum 7021-1.

A	m1	m2	m3	m4
0.00025360974	−0.00980421670	−0.064640867300	−0.20577738210	−0.00794888640
**m5**	**m7**	**m8**	**m9**	
0.00005917045	0.09372048366	0.000695671530	2.67141264040	

### 3.2. Analysis of Plastometric Test Results and Development of Material Model Coefficients of Aluminum 7021-2

On the basis of preliminary tests conducted for aluminum melt 7021-2, it was found that during strain at 570 °C (Figure 5f), regardless of the strain rate used, the material lost cohesion at the beginning of the strain process, which was characterized by a sharp decrease in the value of the flow stress. Accordingly, it was decided to modify the scope of the plastometric tests by lowering the temperature from 570 °C to 560 °C. Due to further loss of cohesion of the material during strain at 560 °C, the temperature was lowered to 550 °C in the next stage of testing. Loss of 7021-2 melt cohesion at 560 °C and 570 °C is related to the solidus temperature. In the case of the 7021-2 melt after the homogenization process, the solidus temperature was 572.1 °C [29]; therefore, this aluminum melt lost cohesion during deformation at a temperatures of 560 °C and 570 °C, which are close to the solidus temperature.

The true and approximated courses of changes in the flow stress of aluminum 7021-2, over the range of temperature and strain rate studied, are shown in Figure 5.

After analyzing the plastic flow curves of the studied 7021-2 aluminum melt, deformed at 450 °C, it was found that during strain at 0.05 s^−1^ and 5 s^−1^, the true values of flow stress (blank symbols) remained constant after reaching the maximum value. The observed slight increase in stress values at the final stage of the strain process may be due to friction at the interface between the surface of the material being deformed and the tool (despite the use of graphite lubricant and graphite film). On the other hand, when straining the tested material at a speed of 0.5 s^−1^, the true values of the flow stress (blank symbols) showed a monotonic increase in its value with rising strain. As a result of the increase in strain rate from 0.05 s^−1^ to 0.5 s^−1^, the true flow stress of aluminum 7021-2 increased (on average) by about 54%. In contrast, an increase in strain rate from 0.5 s^−1^ to 5 s^−1^ increased the true level of flow stress (on average) by about 20%.

Based on an analysis of the data shown in Figure 5b, it was found that during the deformation of the aluminum melt tested at 480 °C at strain rates of 0.05 s^−1^ and 5 s^−1^, the true values of flow stress (blank symbols) remained constant after reaching a maximum value (as in the case of strain at 450 °C). On the other hand, when the material was deformed at 480 °C with a strain rate of 0.5 s^−1^, a monotonic increase in the true values of flow stress was observed again as the strain rate increased. As a result of the increase in strain rate from 0.05 s^−1^ to 0.5 s^−1^, the true flow stress of aluminum 7021-2 increased at that temperature (on average) by about 67%. In turn, an increase in strain rate from 0.5 s^−1^ to 5 s^−1^ increased the true level of flow stress (on average) by about 28%.

A similar nature of the course of the true values of the flow stress (blank symbols) was observed during the strain of the alloy at 510 °C (Figure 5c) and 540 °C (Figure 5d). Based on an analysis of the test results shown in Figure 5c, it was found that as the strain rate increased from 0.05 s^−1^ to 0.5 s^−1^, the true flow stress of aluminum 7021-2 increased (on average) by about 71%. In turn, an increase in strain rate from 0.5 s^−1^ to 5 s^−1^ increased the true level of flow stress of the tested material by about 24% (on average). When 7021-2 aluminum was deformed at 540 °C (Figure 5d), increasing the strain rate from 0.05 s^−1^ to 0.5 s^−1^ increased the true flow stress (blank symbols) of 7021-2 aluminum (on average) by about 80%. In contrast, an increase in strain rate from 0.5 s^−1^ to 5 s^−1^ increased the true level of flow stress (on average) of the tested material by about 22%.

After analyzing the graphs of changes in the true flow stress (blank symbols) of the 7021-2 melt deformed at 550 °C (Figure 5e) at a strain rate of 0.05 s^−1^, it was found that (similar to the strain of the analyzed material in the temperature range 450 °C–540 °C) after reaching the maximum value, the true values of the flow stress of the analyzed melt remained constant. Analyzing the data shown in Figure 5e, it was found that when the material was deformed at 550 °C at 0.5 s^−1^ and 5 s^−1^, the nature of the true plastic flow curves of aluminum 7021-2 changed. When the test melt was deformed at a strain rate of 0.5 s^−1^, the true flow stress (blank symbols) remained constant after reaching the maximum value. On the other hand, when straining the tested material at a rate of 5 s^−1^, after reaching the maximum value, the true values of the flow stress were slightly reduced, which may indicate the occurrence of the recrystallization process in the tested aluminum alloy.

When 7021-2 aluminum was deformed at 550 °C (Figure 5e), increasing the strain rate from 0.05 s^−1^ to 0.5 s^−1^ increased the true flow stress (on average) by about 65%. In contrast, an increase in strain rate from 0.5 s^−1^ to 5 s^−1^ increased the true level of flow stress (on average) of the tested melt by about 61%.

Analyzing the effect of temperature in the range of 450 °C–540 °C on the level of true values of the flow stress of aluminum 7021-2 (blank symbols) deformed at 0.05 s^−1^ and 0.5 s^−1^, it was found that the true values of flow stress decreased (on average) by ca. 56%. When the test melt was deformed at a rate of 0.5 s^−1^, the true values of flow stress were found to decrease (on average) by ca. 53% over the temperature range studied. In turn, when the test melt was deformed at a rate of 5 s^−1^, the true values of flow stress were found to decrease (on average) by ca. 38% over the temperature range studied.

For the analyzed aluminum melt, the limiting temperature at which the material did not lose cohesion during the strain process was 550 °C. Table 3 shows the values of the coefficients of the Hensel–Spittel equation (Equation (4)), which approximates the results of plastometric tests of aluminum 7021-2.

**Table 3 materials-18-03166-t003:** Material model coefficients of aluminum 7021-2.

A	m1	m2	m3	m4
587403094.381	−0.00052313060	−0.06090832670	0.04764237665	−0.00936113470
**m5**	**m7**	**m8**	**m9**	
−0.00068964580	0.38286445658	0.00022502114	−2.61072463990	

After analyzing the course of the true and approximated plastic flow curves of aluminum 7021-2 in the studied range of strain parameters, one can observe a high correspondence between the true values of the flow stress and the values obtained by approximation.

### 3.3. Analysis of Plastometric Test Results and Development of Material Model Coefficients of Aluminum 7021-3

On the basis of preliminary plastometric tests conducted for aluminum 7021-3, it was found that during strain at 570 °C (Figure 6f), regardless of the strain rate used, the material (as in the case of melt 7021-2) lost cohesion at the beginning of the strain process, which was characterized by a sharp decrease in the value of the flow stress. Accordingly (as in the case of melt 7021-2), the decision was made to modify the scope of the tests by lowering the temperature from 570 °C to 560 °C. Due to the loss of cohesion of the melt under test during strain at 560 °C, the temperature was lowered to 550 °C in a subsequent stage of plastometric testing.

The true and approximated courses of changes in the flow stress of aluminum 7021-3, over the range of temperature and strain rate studied, are shown in Figure 6.

On the basis of the analysis of the plastic flow curves of aluminum 7021-3, deformed at 450 °C, it was found that during strain at 0.05 s^−1^ and 5 s^−1^, the true values of flow stress (blank symbols) after reaching the maximum value remained constant, which may indicate the occurrence of healing processes in the structure of the material tested under these conditions. On the other hand, when straining the tested material at a speed of 0.5 s^−1^, the true values of the flow stress of the tested melt showed a monotonic increase in value with rising strain. As a result of the increase in strain rate from 0.05 s^−1^ to 0.5 s^−1^, the true flow stress of aluminum 7021-3 increased (on average) by about 57%. In contrast, an increase in strain rate from 0.5 s^−1^ to 5 s^−1^ increased the level of flow stress (on average) by about 24%.

After analyzing the test results shown in Figure 6b, it was found that during the strain of the aluminum melt tested at 480 °C with a strain rate of 0.05 s^−1^, the true values of flow stress (blank symbols) remained practically constant after reaching the maximum value. On the other hand, when the material was deformed at 480 °C with a strain rate of 0.5 s^−1^, a monotonic increase in the true values of flow stress was observed as the strain rate increased. In turn, when the tested material was deformed at 480 °C with a strain rate of 5 s^−1^, the true flow stress of the tested melt showed a slight monotonic decrease in value after reaching the maximum value. The observed slight increase in true stress values at the final stage of the strain process may be due to friction at the interface between the surface of the material being deformed and the tool (despite the use of graphite lubricant and graphite film). As a result of the increase in strain rate from 0.05 s^−1^ to 0.5 s^−1^, the true flow stress of aluminum 7021-3 increased at that temperature (on average) by about 45%. In contrast, an increase in strain rate from 0.5 s^−1^ to 5 s^−1^ increased the true level of flow stress (on average) by about 33%.

Based on the analysis of changes in the flow stress of aluminum 7021-3 deformed at 510 °C (Figure 6c), it was found that when the material was deformed at strain rates of 0.05 s^−1^ and 0.5 s^−1^, the true values of the flow stress (blank symbols) showed a slight increase with increasing strain values. On the other hand, when the material was deformed at a strain rate of 5 s^−1^, the true flow stress of aluminum 7021-3 remained constant after reaching the maximum value, which may indicate that healing processes were taking place in the material. As a result of the increase in strain rate from 0.05 s^−1^ to 0.5 s^−1^, the true flow stress of aluminum 7021-3 increased at that temperature (on average) by about 64%. In contrast, an increase in strain rate from 0.5 s^−1^ to 5 s^−1^ increased the true level of flow stress (on average) by about 32%.

After analyzing graphs of changes in the true values of the flow stress (blank symbols) of aluminum 7021-3 deformed at 540 °C (Figure 6d) at strain rates of 0.05 s^−1^ and 0.5 s^−1^, its value was found to increase minimally as the strain rate increased. On the basis of changes in the true values of the flow stress of aluminum 7021-3 deformed at 540 °C with a strain rate of 5 s^−1^, it was found that after reaching the maximum value, the true stress monotonically decreased in value with an increase in the value of the set strain, which may indicate the occurrence of a recrystallization process in the structure of the material.

After analysis of the effect of temperature in the range of 450 °C–540 °C on the level of true values of flow stress (blank symbols) of 7021-3 aluminum deformed at 0.05 s^−1^ and 0.5 s^−1^, it was found that the true values of flow stress decreased (on average) by ca. 48-49%. When the test melt was deformed at a rate of 5 s^−1^, the true values of flow stress were found to decrease (on average) by ca. 38% over the temperature range studied. On this basis, it was found that the aluminum smelting studied was sensitive to both an increase in strain rate and a change in temperature.

As can be seen from the data shown in Figure 6e, regardless of the applied strain rate when straining the material at 550 °C, the graphs show a sharp decrease in the value of the flow stress at the beginning of the strain process due to the loss of cohesion of the aluminum alloy tested. Loss of 7021-3 melt cohesion in the temperature range of 550 °C–570 °C is related to the solidus temperature. In the case of the 7021-3 melt after the homogenization process, the solidus temperature was 559.2 °C [29]; therefore, this aluminum melt lost cohesion during deformation at a temperature of 550 °C, which is close to the solidus temperature. For aluminum grade 7021-3, the limiting temperature at which the material did not lose cohesion during the strain process was 540 °C.

Table 4 shows the values of the coefficients of the Hensel–Spittel Equation (4), which approximates the results of plastometric tests of aluminum 7021-3.

**Table 4 materials-18-03166-t004:** Material model coefficients of aluminum 7021-3.

A	m1	m2	m3	m4
16213370673329	0.00394792524	−0.03027832910	−0.12121689290	−0.00866636950
**m5**	**m7**	**m8**	**m9**	
−0.00162790580	0.50829874675	0.00056570951	−4.61264296720	

After analyzing the course of the true and approximated plastic flow curves of aluminum 7021-3 in the studied range of strain parameters, one can observe a high correspondence between the true values of the flow stress of the analyzed melt and the values obtained by approximation.

### 3.4. Results of Numerical Modeling of the Extrusion Process

The immediate results of the study are graphical visualizations of the way the metal flows out of the die cavity, shown in Figure 7 and Figure 8 for three different melts (using dedicated material models) extruded using the same porthole die. The distribution of the analyzed parameter of the variation in the metal discharge velocity in the die clearance with respect to the mean value (velocity deviation) is presented in the form of a multi-color scale. First of all, the variation in metal plastic melt velocity in the die clearance cross-section was analyzed at the process initiation stage—Step 1 (Figure 7). The parameter values were also determined in characteristic localizations where the way the metal flows varies and is determined by the die design.

Measuring points labeled No. 1 are located in the central part of the die and in the area of direct feeding of bearing from the billet, so the intensity and speed of metal flow should be the highest in this area, which is evident in the case of melts 7021-2 (b) and 7021-3 (c). In the case of the 7021-2 variant, this value is 7.6%, while in the case of the 7021-3 variant, it is as high as 43.82%, which is unacceptable from the point of view of the correctness of the process parameters and the quality of the extrudates themselves. It can be concluded that description of the behavior of the metal and the tendency of the way it flows is correct, but significantly amplified. In the case of alloy 7021-1, the trend is quite different, i.e., negative values of −4.9% are observed, which may indicate a lower susceptibility to plastic flow of the material under analogous process conditions for a given die design. Measuring points No. 2 and No. 4 are located, in turn, in the die clearance located under the welding chambers in the area of half of the billet feed radius, which should translate into significantly lower values of metal flow in the die clearance. This effect is seen especially in the case of values of measuring points labeled No. 2—negative values are observed for all melts, while for alloy 7021-3 (c) these values are higher by more than a half, indicating a much lower intensity of metal plastic flow.

Figure 8 shows the results of numerical FEM simulations in a stabilized state, i.e., in the 10th step of the simulation—where the way profiles flow out of the die cavity is already fully developed. This provides an opportunity to assess the tendency for variation in metal outflow from the bearing and to visually assess the discrepancy between the different 7021 melts as well as the deviation from the desired geometry.

It can be observed that in each of the cases in Figure 8, there is a trend to increase the strain caused by local unevenness of metal outflow from bearings conditioned by the die design.

In the case of Figure 8a, a much slower inflow of metal is observed in the central part, which is not fully consistent with the natural tendency of metal to flow in the extrusion process. According to the theory of extrusion processes, the intensity of flowing decreases on the cross-sectional diameter of the billet in the direction of its outer edges due to the frictional forces of the billet against the press container. The highest metal flow velocity in this case is observed in the outer areas of the die, which may indicate the inadequate suitability of the melt and material model 7021-1 for the extrusion process of the sections in question and the different way the metal flows through the die. The geometry of the sections is clearly deformed relative to the nominal section, which can cause both lack of centricity of the tubes and variation in wall thickness. In addition, it creates technological problems related to the proper localization of extrudates on the press and placement in the puller.

In the case of the melt and material model 7021-2, the effect of intensification of metal flow in the central part is clearly visible, which translates into strain and tendency for outflow in the external direction of the die, which is manageable from the point of view of the technological aspect of the production process. In the case of the melt and material model 7021-3, a similar intensification effect is observed in the central area, but the value of velocity deviation significantly exceeds ±10%, which clearly disqualifies the tested technology from implementation. The use of this melt in the industrial process of extrusion using a proprietary die is pointless due to potential technological difficulties and the expected quality of extrudates. The visual manner of extrudate strain determined in numerical simulations, conditioned by unevenness from the bearing, also makes it clear that, in the case of melts and material models 7021-1 and 7021-3, the geometric accuracy of the final products will not meet the standard of acceptable dimensional tolerances.

To determine the force parameters of the process, numerical simulations were carried out for the whole billet simulation. Thus, we determined the change in the value of the extrusion force as a function of the punch path corresponding to the length of the extruded billet (800 mm long). Based on the data obtained, comparative curves were developed for melts and material models 7021-1, 7021-2 and 7021-3, as shown in Figure 9.

For each variant, one simulation was conducted due to constant parameters calculated for material model and the time-consuming nature of the process. As can be seen in the graph, the nature of the curves is similar and the main differences are manifested in the value of the maximum force of initiation of the extrusion process. The value of punch displacement corresponds in each case to 110 mm, when the metal has completely swollen the interior of the die reaching the die clearance, starting the extrudate forming process. The discrepancies in the values of Fmax (maximum force of process initiation) are a direct result of the rheological properties (material models) of the materials tested. In conjunction with the previously analyzed method of metal flow out of the die cavity, this creates the possibility of selecting and determining the most attractive melt and material model for production. Therefore, these two parameters (product shape and extrusion force) should be analyzed simultaneously. In the case of the melt and material model 7021-1, the value of the maximum force is the lowest out of the entire group of materials. It does not exceed 15 MN, which is quite low, but beneficial from the point of view of energy. Analysis of the outflow pattern shown in Figure 8 indicates the incorrect tendency of metal flowing in the extrusion process through the porthole die, which may indicate, among other things, an incorrect approximation of the rheological properties of the material.

In the case of the melt and material model 7021-2, the value of the maximum process initiation force is about 21 MN, which is the correct value at least from a theoretical point of view, and it is also within the force capabilities of the press. The method for metal flowing out of the die cavity determined by numerical simulations also does not raise theoretical concerns. Based on the results, this melt (together with the material model) was selected for further validation studies under industrial conditions.

The maximum force value for the start of the extrusion process for the melt and material model 7021-3 is 35 MN, significantly exceeding the force capabilities of an industrial press. Also, the way the metal flows out of the die cavity indicates that the resistance to plastic flow of the metal is too high. Therefore, the melt (together with the material model) was rejected from further validation studies under industrial conditions.

### 3.5. Results of Industrial Verification

Conducting numerical simulations of the extrusion process using a standard die under conditions analogous to industrial trials made it possible to plot the course of change in the calculated extrusion force as a function of the punch path during billet extrusion. The results obtained were compared with experimental data obtained during industrial tests using the proprietary die made according to the developed design shown in Figure 10a,b. This determined the accuracy of numerical calculations by comparing process parameters and evaluating how the initial section of the profile flows out on the press (Figure 10c).

The values of the pressure recorded in the hydraulic system of the press and the calculated value of the force are summarized on the vertical axes. The summary was prepared by determining the maximum value of pressure in the press system at 250 Bar and the corresponding maximum force of 25 MN defined in the press documentation.

The developed dependencies (results) obtained in industrial tests and in numerical simulations are shown as graphs in Figure 11. The vertical axes show the pressure values recorded in the press hydraulic system and the calculated force values. The summary was prepared by setting the maximum pressure in the press system at 250 bar and the corresponding maximum force defined in the press documentation at 25 MN. The graph shows the extrusion pressure as a function of ram displacement for one representative billet and corresponding simulation. The results of the remaining trials were consistent with each other.

Both the maximum value of the Fmax force in the process and the displacement of the punch at which this parameter was recorded were analyzed. The curves are presented in a dual system represented by the maximum pressure value in the hydraulic system of the industrial press at 250 bar and simultaneously in the Extrusion Load system of the press under virtual conditions defined according to actual conditions at 25 MN.

The nature and course of the curve during the process were evaluated. Analysis of the plotted relationships allows us to conclude that the simulation results present the prevailing conditions in the process with high accuracy. This is evidenced primarily by the maximum value of the force Fmax of process initiation. The calculated value amounts to 21 MN, while the value determined in industrial tests is 23 MN; thus, the difference does not exceed 10%, which confirms the accuracy of the simulation (i.e., the high accuracy of the material model). The difference may be due to imperfect industrial conditions for performing the tests, i.e., uneven die heating, billet or increased friction due to efficiency of the press. Also, the value of stamp displacement at which the maximum value was recorded is close to 110 mm, which shows that the process was accurately mapped in terms of both describing the rheological properties and the geometry of the toolset. This value deviates slightly by less than 25 mm from the calculated one, which, among others, may be due to billet cutting precision or leaking of contact of the dummy block with the container in the extrusion press.

Figure 12 shows a comparison of the way the initial part of the nose piece extrudate is deformed, arriving at an almost identical trend between the results of numerical simulations and the actual flow of metal out of the die cavity, taking into account the location of the holes in the tool set itself. Both the profiles from cavity 1 and cavity 2 provide an outflow pattern consistent with that determined in the simulations. The area located in the center flows out much faster, while in the outer area the aluminum moves much slower, especially in the initial stage of die filling. The results confirm the high accuracy of numerical simulations using a dedicated material model of melt 7021-2. The geometric accuracy was investigated in a comprehensive survey using a photogrammetric system.

#### Geometric Accuracy Testing

Within the framework of the geometric tests carried out on the tubes, special attention was paid to the analysis of the compliance of the obtained measurement results with the requirements of EN 755-8 [39], which defines the permissible dimensional tolerances for extruded products made of aluminum alloys.

The analysis focused on evaluating three key geometric parameters: inner diameter, outer diameter and wall thickness. Measurements were made using high-precision 3D scanning, which allowed for a detailed representation of the geometric profile of the components studied and made it possible to identify any deviations from nominal values.

The obtained values were then compared with the permissible tolerances specified in EN 755-8, which allowed an objective assessment of the geometric quality of the tested tubes in the context of the applicable technical standards.

Figure 13 shows the results of the internal and external diameters for the geometric variant of the tube analyzed. The values obtained indicate high accuracy of the component, which confirms compliance with the requirements of EN 755-8.

The dimensional deviations discovered are within the permissible tolerance range specified in this standard. The maximum deviation for the outer diameter was 0.12 mm, while for the inner diameter it was 0.16 mm, indicating stability of the production process and precision of manufacturing.

Figure 14 shows the results of wall thickness measurements for the variants of extruded profiles. Based on wall thickness measurements, it was found that all the analyzed values were within the permissible tolerances specified in EN 755-8. The results obtained confirm the geometric compliance of the components with the normative requirements, which indicates proper control of the technological process.

The recorded wall thickness deviations from the nominal value ranged from −0.8 mm to +0.5 mm, which means that the maximum discrepancies do not exceed the limits set by the applicable standards.

The results of tube wall thickness measurements were statistically analyzed (Table 5) to assess compliance with the requirements of EN 755-8.

Figure 15 shows the distribution of the tube wall thickness values measured at 32 measurement points distributed along its length. The presented result is shown as a representative example for one sample from the series, on which 32 measurements were performed. The results obtained from the remaining samples are also within the acceptable range. A total of 10 profiles were tested. These values were compiled taking into account the tolerance limits specified in EN 755-8, according to which the permissible deviation for wall thickness with a nominal value of 2 mm is ±0.35 mm, which corresponds to a range of 1.65 mm to 2.35 mm.

All the results obtained are within the designated tolerance range, which confirms the high quality of workmanship and the stability of the technological process. The slight variations in thickness seen in the graph do not exceed the maximum values allowed by the standard, and their nature does not indicate significant disturbances in the homogeneity of the material or technological deformations.

## 4. Discussion

Aluminum alloys of the 7xxx series are hard and durable aluminum alloys with zinc and magnesium. This alloy has the highest strength of any aluminum alloys—comparable to structural steels.

Among others, the effect of the applied strain parameters on the rheological properties of the three studied aluminum alloy melts in the EN AW-7021 grade was determined on the basis of plastometric tests and after analyzing their results. In addition, the tests showed that the plasticity of the three EN AW-7021 aluminum melts analyzed depended on the chemical composition. Of the melts tested, melt 7021-1 had the best plasticity. The material showed no signs of cohesion loss over the entire range of temperature and strain rate tested. The lowest plasticity was demonstrated by melt 7021-3, whose limiting temperature at which the material did not lose cohesion during the strain process was 540 °C. This is directly related to the different contents of Zn and Mg in the analyzed melts. In the studied range of strain parameters, an increase in Zn content combined with an increase in Mg content resulted in a decrease in the plasticity of the analyzed EN AW-7021 aluminum melts.

The slight increase in the true values of flow stress at the final stage of the strain process, observed during plastometric tests, may, according to the authors of the paper, be due to friction at the interface between the surface of the deformed material and the tool (despite the use of graphite lubricant and graphite film).

The developed dedicated material models for aluminum alloy 7021 were implemented using the Hensel–Spittel constitutive equation, which made it possible to comprehensively define and approximate the rheological properties of the material during the industrial extrusion process. It was found that the selected constitutive equation provides a comprehensive definition of the material over a wide range of strain parameters.

Numerical simulations carried out for industrial parameters, where the variable (dedicated) parameter was only a different material model, allowed us to accurately determine its influence on the mode of strain and plastic flow of metal. It was found that the accuracy of the material model translates into the geometric accuracy of extrudates and the force parameters of the process.

Validation of the results obtained in the numerical simulations made it possible to determine the degree of accuracy of the developed material model. It was found that for the tested melt and material model 7021-2, the maximum value of the extrusion force determined under industrial conditions is only slightly higher than the values determined in simulations, and does not exceed the value of 10%. Also, the course of the curve and the position of the punch at which the maximum extrusion force value occurs are similar, which shows the high accuracy of results of numerical simulations.

Based on the analysis of simulation results, a melt was selected for verification testing in industrial trials using a proprietary die melt (and material model); 7021-2 was found to have the most uniform metal flow through the selected die and to meet the maximum process initiation force requirements on the selected 25 MN press. The analysis of the way the metal flows made it possible to conclude that the deviation from the mean value (velocity deviation) for a given variant does not exceed ±8% throughout the bearing, ensuring high dimensional accuracy of the extrudate, which was confirmed by optical scanning tests.

## 5. Conclusions

After conducting tests related to the development and verification of dedicated material models of the EN AW-7021 alloy for numerical modeling of the industrial process of extrusion of profiles of hard-strain aluminum alloys through porthole dies, the following conclusions were formulated after analyzing the obtained results:Plastometric tests conducted on three analyzed aluminum melts of the EN AW-7021 grade showed significant effects of strain, temperature and strain rate on flow stress values.For all analyzed EN AW-7021 aluminum melts, as the strain rate increases, there is a concomitant increase in the value of flow stress, within the investigated range of strain parameters.An increase in the temperature of the EN AW-7021 aluminum melts tested results in a decrease in the value of the flow stress (for the corresponding strain rates).For each developed EN AW-7021 aluminum material model, by analyzing the course of the true and approximated plastic flow curves, a significant correspondence can be observed between the true values of flow stress and those obtained by approximation.In the case of melt 7021-2, the limiting temperature at which the material did not lose cohesion during plastometric testing was 550 °C.For melt 7021-3, the limiting temperature at which the material did not lose cohesion during the strain process was 540 °C.Taking into account the actual rheological properties of the three analyzed EN AW-7021 aluminum melts during the numerical modeling of the extrusion process provided an increase in the accuracy of calculations with respect to the actual technological process.Conducting numerical simulations using the developed material models allows us to select the most favorable model for the extrusion process for the designed technology using the proprietary die. This makes it possible to reduce the number of industrial tests and fully control the parameters and quality of the products.The developed material model and its detailed description by means of the constitutive equation translate directly into the quality of the results of numerical simulations, which is confirmed both by the analysis of the way metal flows in the extrusion process using the proprietary die and the force characteristics confronted with the results obtained in industrial tests.Tests of the quality of extruded profiles confirmed the high geometric accuracy of extruded products under stable conditions in accordance with the relevant standard EN 755-8.

## Figures and Tables

**Figure 1 materials-18-03166-f001:**
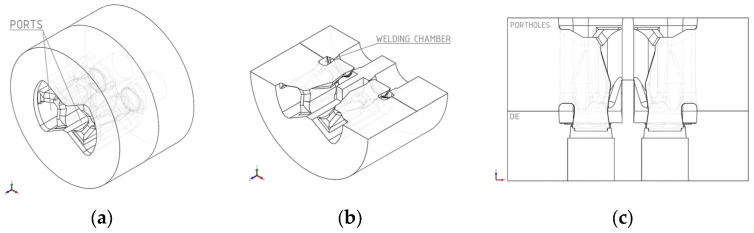
Structure of the proprietary 2-cavity porthole die: (**a**) three-dimensional isometric projection; (**b**) cross-section relative to the horizontal axis of symmetry—isometric projection; (**c**) cross-section relative to the horizontal axis of symmetry—rectangular projection.

**Figure 2 materials-18-03166-f002:**
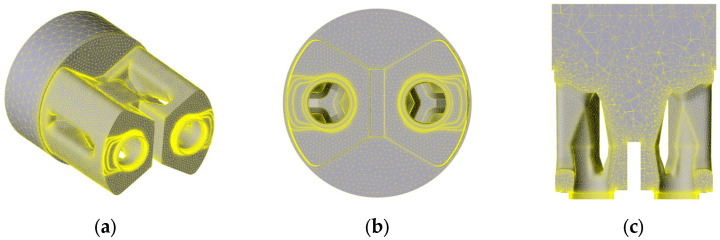
Finite element mesh of the computational domain (aluminum filling the interior of the die): (**a**) isometric view including bearing, pre-chambers and ports; (**b**) lay-out view of the die; (**c**) cross-sectional view of the computational domain filling the die including welding chambers, pre-chambers and bearings.

**Figure 3 materials-18-03166-f003:**
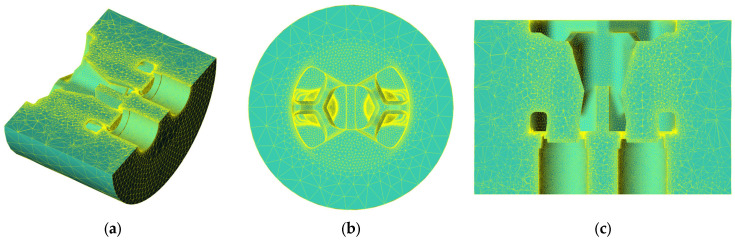
Finite element mesh including areas of compaction: (**a**) isometric cross-sectional view; (**b**) port entrance view; (**c**) cross-sectional view including cores and welding chambers.

**Figure 4 materials-18-03166-f004:**
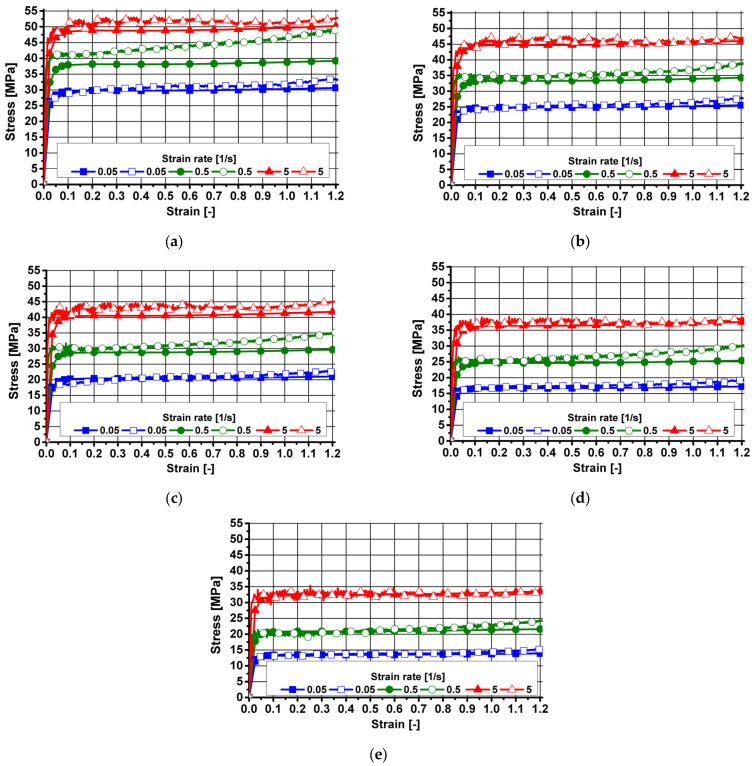
Plastic flow curves of aluminum 7021-1: (**a**) temperature: 450 °C; (**b**) temperature: 480 °C; (**c**) temperature: 510 °C; (**d**) temperature: 540 °C; (**e**) temperature: 570 °C; blank symbols—experimental data; full symbols—results after approximation.

**Figure 5 materials-18-03166-f005:**
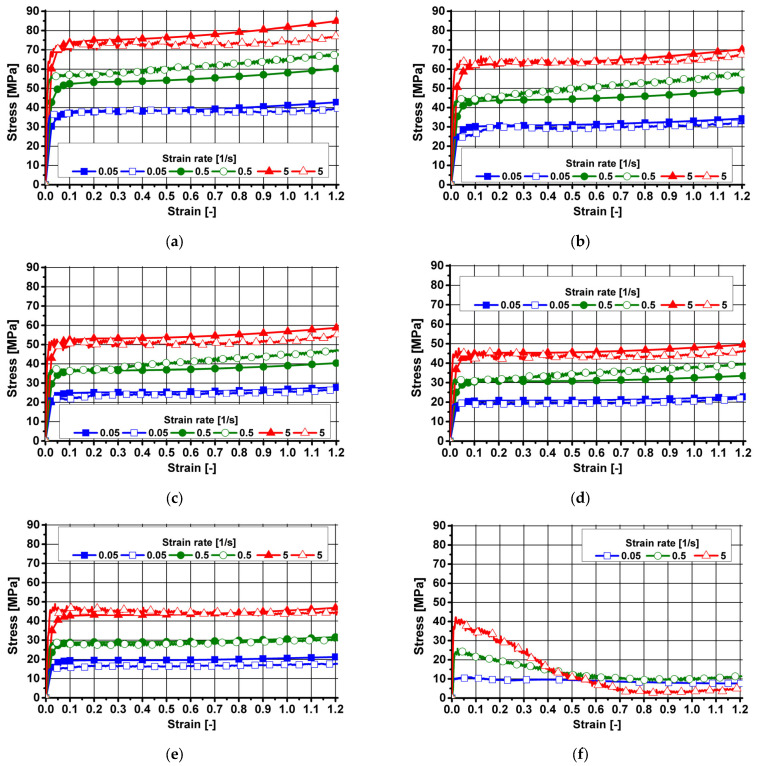
Plastic flow curves of aluminum 7021-2: (**a**) temperature: 450 °C; (**b**) temperature: 480 °C; (**c**) temperature: 510 °C; (**d**) temperature: 540 °C; (**e**) temperature: 550 °C; (**f**) temperature: 570 °C; blank symbols—experimental data; full symbols—results after approximation.

**Figure 6 materials-18-03166-f006:**
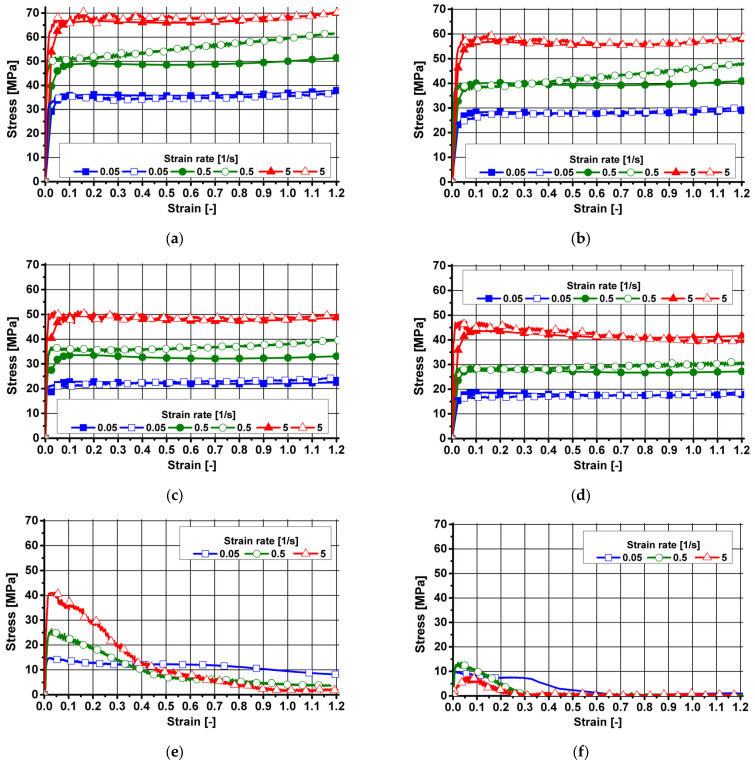
Plastic flow curves of aluminum 7021-3: (**a**) temperature: 450 °C; (**b**) temperature: 480 °C; (**c**) temperature: 510 °C; (**d**) temperature: 540 °C; (**e**) temperature: 550 °C; (**f**) temperature: 570 °C; blank symbols—experimental data; full symbols—results after approximation.

**Figure 7 materials-18-03166-f007:**
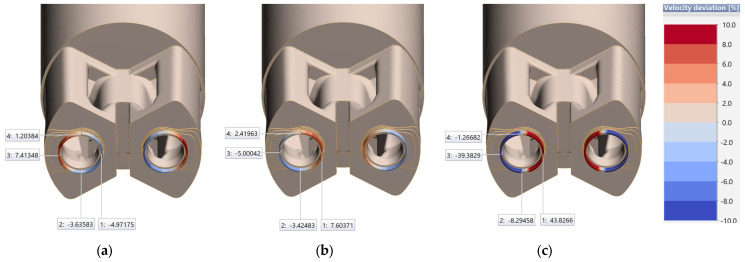
Variation in metal discharge from the die cavity at the initial stage of initiation of the extrusion process by the proprietary die for the rest of the melts: (**a**) 7021-1, (**b**) 7021-2 and (**c**) 7021-3.

**Figure 8 materials-18-03166-f008:**
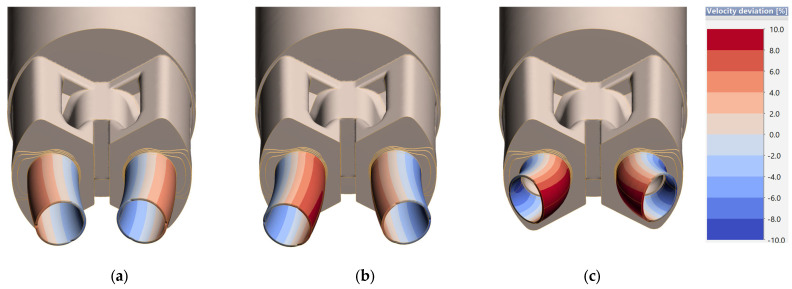
Variation in metal discharge from the die cavity at the stabilized stage of the extrusion process by the proprietary die for the rest of the melts: (**a**) 7021-1, (**b**) 7021-2 and (**c**) 7021-3.

**Figure 9 materials-18-03166-f009:**
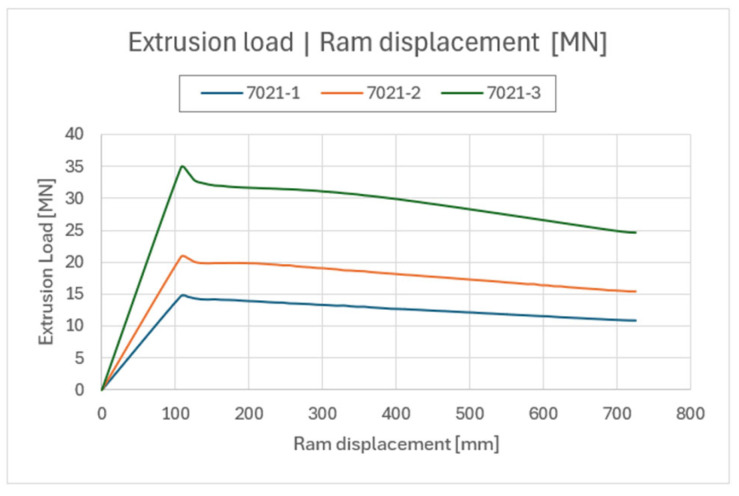
Comparative curves of press force parameters for aluminum melts 7021-1, 7021-2 and 7021-3.

**Figure 10 materials-18-03166-f010:**
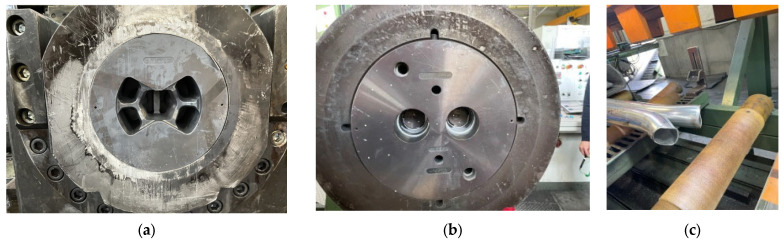
Documentation of industrial tests: (**a**,**b**) die placed in the press; (**c**) initial fragment of profiles on the press.

**Figure 11 materials-18-03166-f011:**
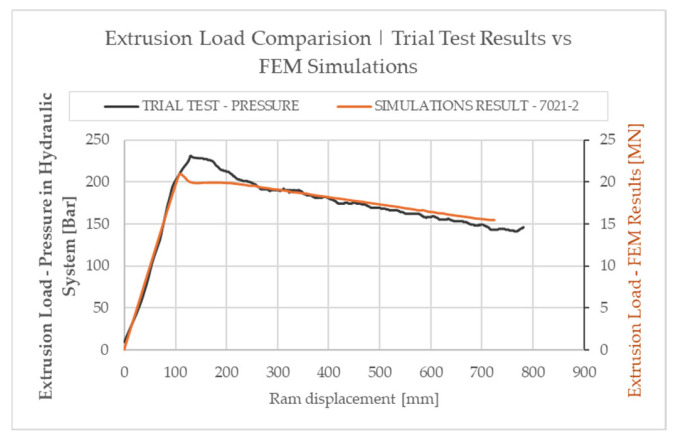
Comparative plot of force parameters of the extrusion process as a function of the punch path recorded during trial testing against the values determined by numerical simulations.

**Figure 12 materials-18-03166-f012:**
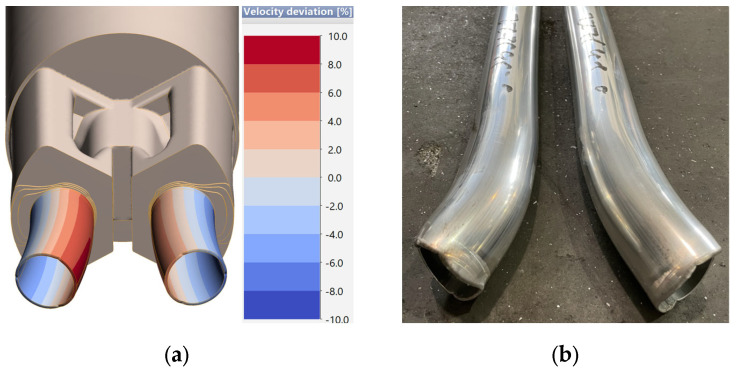
Comparison of the initial part of the nose piece: (**a**) determined in numerical simulations; (**b**) in the actual trial test under industrial conditions.

**Figure 13 materials-18-03166-f013:**
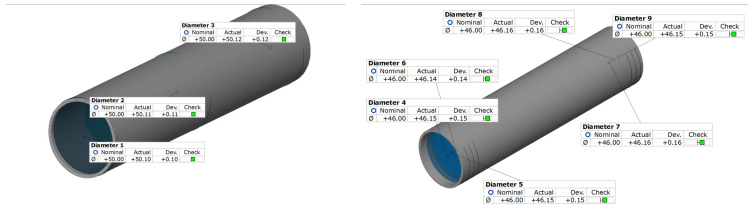
Diameter measurement results for the Ø50 × 2 mm tube from melt 7021, extruded with the proprietary die obtained using 3D optical scanning.

**Figure 14 materials-18-03166-f014:**
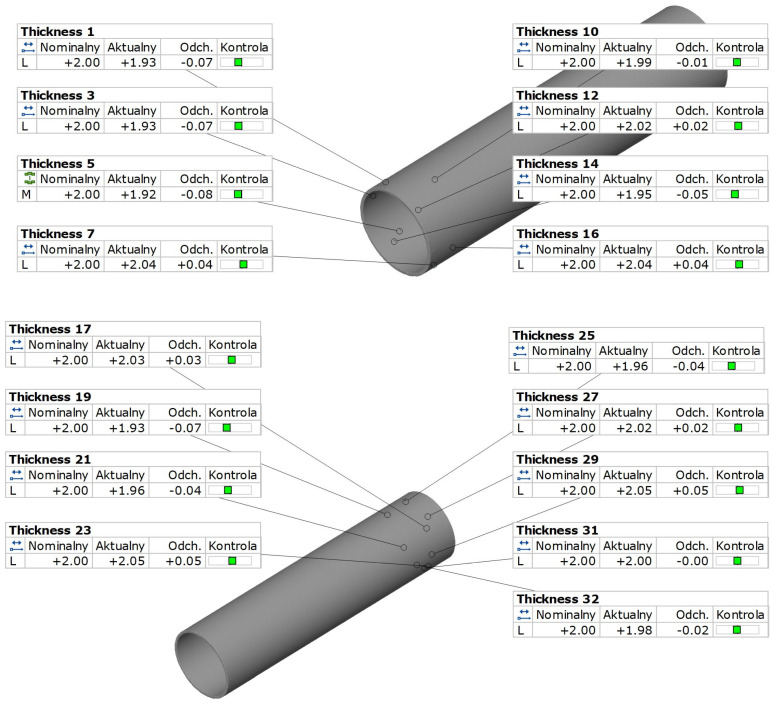
Wall thickness measurement results for the Ø50 × 2 mm tube from melt 7021, extruded with the proprietary die obtained using 3D optical scanning.

**Figure 15 materials-18-03166-f015:**
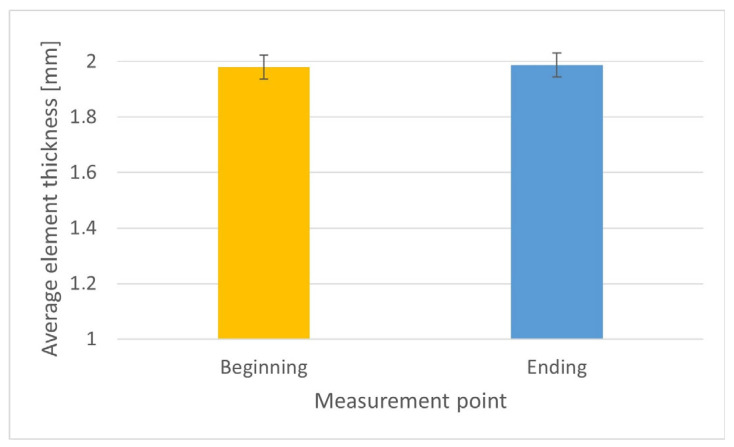
Results of wall thickness measurements of extruded tubes with a diameter of 50 mm from melt 7021-2.

**Table 1 materials-18-03166-t001:** Chemical composition of the tested EN AW-7021 aluminum alloy [29].

Melt Designation	Melt Analysis, by % Weight								
	Si	Fe	Cu	Mn	Mg	Cr	Zn	Ti	Zr
7021-1	0.09	0.22	0.00	0.00	**1.20**	0.00	**5.27**	0.01	0.15
7021-2	0.09	0.21	0.00	0.00	**2.12**	0.00	**5.47**	0.01	0.15
7021-3	0.09	0.22	0.01	0.00	**2.06**	0.00	**8.02**	0.02	0.15

**Table 5 materials-18-03166-t005:** Summary results of the analysis of wall thickness measurements of the analyzed extruded profile.

Minimum Tube Wall Thickness, mm	Maximum Tube Wall Thickness, mm	Variation in Tube Wall Thickness, %	EN 755-8 Standard, %	Average Value, mm	Standard Deviation	Median, mm
1.92	2.05	6.50	max ± 7	1.98	0.043	1.985 mm

## Data Availability

The original contributions presented in this study are included in the article. Further inquiries can be directed to the corresponding author.

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
