# Peer review of "Dedicated Material Models of EN AW-7021 Alloy for Numerical Modeling of Industrial Extrusion of Profiles"

_materials, 2025, doi:10.3390/ma18133166_

Round 1
Reviewer 1 Report
Comments and Suggestions for Authors
In this study, the authors represent an in-depth study on the development and verification of dedicated material models for three EN AW-7021 aluminum alloy melts, differing in Zn and Mg content, under conditions representative of industrial extrusion through porthole dies. The authors combine plastometric testing using a GLEEBLE simulator with numerical modeling in QForm, followed by experimental extrusion trials to validate the results. This study provides insights for researchers and engineers involved in aluminum alloy processing and process simulation. While the manuscript is generally well-structured, several concerns need to be addressed before the manuscript can be considered for publication:
- The authors repeatedly conclude that the flow stress of the studied alloy variants is more sensitive to strain rate than to temperature. This conclusion is currently based only on percentage changes in flow stress and lacks a rigorous quantitative basis. Please compute and report standard sensitivity metrics such as the strain rate sensitivity exponent (m) and activation energy (Q) to support this conclusion.
- For melts 7021-2 and 7021-3, the manuscript reports "loss of cohesion" at higher temperatures. Could this behavior be associated with the solidus temperature or onset of incipient melting? Please add a brief discussion of the failure mechanism.
- The Hensel-Spittel model was used to fit the experimental flow stress data. Please state the reason for choosing this specific model and compare it with other constitutive models. Please also provide the goodness of fit (e.g., R² values) for the model fits.
- The manuscript contains several instances of repetitive or redundant phrasing (e.g., “flow stress increased its value”). Simplifying such phrases would improve clarity and readability.
Author Response
Detailed answer in attachment

Reviewer 2 Report
Comments and Suggestions for Authors
Please find the attachment

Author Response
Detailed answer in attachment

Reviewer 3 Report
Comments and Suggestions for Authors
Notes in attachment.

Author Response
Detailed answer in attachment

Reviewer 4 Report
Comments and Suggestions for Authors
This review concerns the manuscript Development and Verification of Dedicated Material Models of Hard-Deformable EN AW-7021 Alloy for Numerical Modeling of Industrial Extrusion of Profiles through Porthole Dies, whose objectives were the development and validation of dedicated material models for the EN AW-7021 alloy, aiming at their application in the numerical modeling of industrial extrusion processes using porthole dies. The article presents a relevant scope and satisfactorily addresses the main proposed topics. However, some minor adjustments are necessary to improve the comprehension and clarity of the manuscript, as detailed below:
- It is not clear whether the phases present after processing were quantified. It is recommended to provide information regarding possible phase changes and, if feasible, their quantification, in order to reinforce the metallurgical conclusions.
- The temperatures employed (450°C to 570°C), even under vacuum, may cause loss of Zn and potentially Mg. Has any analysis been conducted to quantify or mitigate this effect, since the chemical composition can directly impact the rheological results?
- The yield strength obtained appears to be relatively low for this alloy class. The authors are requested to discuss possible causes and compare them with reference data for EN AW-7021-1. For applications that require high yield strength, it is advisable to consider specific heat treatments and consult technical standards, such as EN 755-2, to ensure compliance with design requirements.
- The manuscript correlates the reduction in plasticity with the combined increase of Zn and Mg across the three batches (7021-1, 7021-2, 7021-3). However, the specific metallurgical mechanisms underlying this relationship are not sufficiently detailed. It is suggested that the authors discuss how the variation of these elements affects the microstructure (e.g., formation of intermetallic phases, precipitates, or segregation) and, consequently, plasticity.
- It is recommended to include error bars in the presented graphs (e.g., Fig. 7, Fig. 8, Fig. 9, Fig. 12, Fig. 18), whenever possible, to better illustrate experimental scatter and data robustness.
- Recurring grammatical errors, long constructions, and literal translations that affect the academic fluency in English have been identified. A careful revision of the text is recommended, prioritizing shorter and more objective sentences, and correcting improper use of technical terms, plurals, and prepositions.
Specific examples include:
- “The course of the strengthening curves of the tested materials and the value of flow stress as a function of strain can vary significantly, depending on the test method (compression, tensile or torsion test) and testing equipment.”
Suggestion: Split long sentences into two or more shorter ones. - “the analyzed smelts of the same aluminum alloy were characterized by different sensitivity to strain rate and temperature.”
Correct form: “…showed different sensitivity to strain rate and temperature.” - “…no papers have been found that would demonstrate in detail…”
More natural: “…no papers were found that demonstrate in detail…” - “the geometry of the cross-section of the outflowing section may be distorted.”
Better: “…of the extruded section may be distorted.” - “These articles describes the research methodology…”
Correct: “These articles describe the research methodology…” - “papers [6,8] present the results…”
Correct: “papers [6,8] present results…” (the phrase “the results of” can be omitted for conciseness) - “in paper [9], discusses in detail, among others, the research methodology…”
Better: “Paper [9] discusses in detail the research methodology…” - “on the die of intermetallic phases”
Better: “on the topic of intermetallic phases” - “the finite element mesh moves with the flowing metal to accurately represent the progress of the die filling.”
Better: “…to accurately represent the filling progress of the die.” - “in the studied range of strain parameters, the three smelts of the same aluminum alloy analyzed were characterized by high and different sensitivity to both strain rate and temperature, depending on the content of the main alloying elements.”
Simplify to: “The three smelts analyzed showed varying sensitivity to strain rate and temperature, depending on their main alloying element contents.” - “This enables numerical simulations of the extrusion process using the designed die set for selected aluminum melts..”
Fix double punctuation and rephrase: “This enabled numerical simulations of the extrusion process using the designed die set for the selected aluminum melts.” - Repetitive use of “the test material” and “the tested alloy” can often be replaced by “the material” or “the alloy.”
- “melt” could be replaced by “batch” or “sample” to avoid confusion in technical English.
- “the test melt was strained at a rate of…”
Better: “the sample was deformed at…”
The work is technically relevant and addresses a topic of interest for the field of aluminum alloy forming, but the implementation of the suggestions above is recommended to raise the standard of the manuscript and enhance its scientific contribution.
Comments on the Quality of English LanguageThe English writing in the manuscript is understandable and adequately conveys the technical content, but improvements are needed to meet the standards expected by international journals. Recurring grammatical errors were identified, as well as excessively long sentences, inappropriate use of plurals, prepositions, and literal translations. A careful language revision is recommended, prioritizing shorter sentences, appropriate technical vocabulary, and greater academic fluency in order to improve the clarity, impact, and professionalism of the text.
Author Response
Detailed answer in attachment

Round 2
Reviewer 1 Report
Comments and Suggestions for Authors
The authors have addressed all my previous comments. The current manuscript is ready for acceptance.
Reviewer 2 Report
Comments and Suggestions for Authors
Accepted
Reviewer 4 Report
Comments and Suggestions for Authors
After a detailed analysis of the revised manuscript and the responses provided, I confirm that all reviewer comments have been satisfactorily addressed. The limitations related to the lack of detailed microstructural analysis and compositional control were explicitly acknowledged and discussed, and do not compromise the objectives of the work, given the clearly established focus on rheological modeling applied to the industrial extrusion process. The revised manuscript is clearer, more concise, and aligned with best practices in scientific communication, reflecting significant improvement compared to the previous version.